# Learning and Development of Home Service Robots' Service Cognition Based on a Learning Mechanism

**Fei Lu** *,† , **Min Huang** † , **Xiaolei Li** † , **Guohui Tian** † , **Hao Wu** † and **Wenjia Si** †

School of Control Science and Engineering, Shandong University, Jinan 250061, China;
huangmin@mail.sdu.edu.cn (M.H.); qylxl@sdu.edu.cn (X.L); g.h.tian@sdu.edu.cn (G.T.);
wh911@sdu.edu.cn (H.W.); 201834564@mail.sdu.edu.cn (W.S.)
* Correspondence: lawyerlf@sdu.edu.cn
† These authors contributed equally to this work.

**Abstract:** In order to improve the intelligence of home service robots and resolve their inability to develop service cognition skills in an autonomous, human-like manner, we propose a method for home service robots to learn and develop skills that allow them to perform their services appropriately in a dynamic and uncertain home environment. In a context model built with the support of intelligent sensors and Internet of Things (IoT) technology in a smart home, common-sense information about environmental comfort is recorded into the logical judgment of the robot as a reward provided by the environment. Our approach uses a reinforcement learning algorithm that helps train the robot to provide appropriate services that bring the environment to the user's comfort level. We modified the incremental hierarchical discriminant regression (IHDR) algorithm to construct an IHDR tree from the discrete part of the data in a smart home to store the robot's historical experience for further service cognition. Poor adaptive capacity in a changeable home environment is avoided by additional user guidance, which can be inputted after the decision is made by the IHDR tree. In the early development period, when robots make an inappropriate service decision because they lack historical experience, the user can help fix this decision. Then, the IHDR tree is updated incrementally with fixed decisions to enrich the robot's empirical knowledge and realize the development of its autonomic cognitive ability. The experimental results show that the robot accumulates increasingly more experience over time, and this experience plays an important role in its future service cognition, similar to the process of human mental development.

**Keywords:** home service robots; service cognition; autonomous development; IHDR; reinforcement learning

## 1. Introduction

With the improvement of living standards, people hope that robots will have enough intelligence to actively and correctly identify the services that users need according to current user status and context information. The development of IoT technology makes it easier for robots to access context information [1,2], including the on/off and open/closed status of devices and furniture. At present, the main methods oriented toward the service cognition of home service robots are knowledge-based, learning-based, and "genetic search" approaches. Although these methods have achieved remarkable success in their fields, they have many limitations in robot service cognition. Robots can neither respond flexibly according to a complex and unstructured context using historical experience nor store existing experience in their "brains" like humans. A home service robot's ability to learn from its existing experience and develop its "mind" like a human is key for the realization of real intelligence in robots.

In recent years, many studies have been carried out on the topic of autonomic service cognition in home service robots. Lu et al. [3] proposed the management of information in a smart home using ontology technology. From the constructed ontology model, the autonomous cognition of service tasks can be realized by using the JSHOP2 planner and inference rule base to build the service task. During service cognition, the user's emotions should also be considered. On the basis of the ontology model in [3], Lu et al. [4] proposed a user emotion-based service recognition system in which the user's emotion serves as reward feedback to dynamically adjust the service. Soar, a cognitive service robot presented by Dang et al. [5], constructs a healthcare table as well as a basic profile of each family member and provides the service with the maximum utility value. Aiming to solve the problem of robotic services provided to multiple users at the same time, Lee et al. [6] proposed a scheduling policy to provide the service to multiple users according to a service queue, which is generated according to user intentions. However, the service cognition methods mentioned above are all based on specific tasks, and the developer needs to rebuild and train the model if the task or model changes. This process results in robots that have low intelligence and universality.

Robots can be truly intelligent only if they have the ability to learn by themselves and develop their "minds" autonomously. Reinforcement learning lies in self-learning, and it has been widely used in the field of robotics. In order to provide users with more suitable services, Moon et al. [7] proposed a network-based recommendation robot that could learn users' habits by interacting with them through reinforcement learning, thereby providing personalized services. Focusing on multi-robot systems, Tian et al. [8] proposed a multi-robot task allocation algorithm for fire-disaster response based on reinforcement learning and achieved distributed task allocation for multi-robot systems. To address the problem of indoor localization, Mehdi et al. [9] proposed a model that could closely estimate target locations using deep reinforcement learning (DRL) in IoT and smart city scenarios. Savaglio et al. [10] used reinforcement learning to design an energy-preserving Media Access Control (MAC) protocol, thereby realizing high-density communication in wireless sensor networks (WSN). They established that the application of reinforcement learning in a smart home is technically feasible, which provides the basis for this article. In the field of robotics, reinforcement learning is often used in navigation and path planning [11–13]. Therefore, we focused on applying reinforcement learning to the service cognition of robots in the home environment to provide robots with the ability to learn the skill of service cognition by themselves.

Furthermore, inspired by the learning process of the human brain, we focused on the autonomous development of robots that, controlled by their intrinsic developmental program, develop their mental capabilities through autonomous real-time interactions with environments using their own sensors and effectors [14]. The concept of autonomously developed robots has led to extensive research by experts in related fields. Weng et al. [15] presented an IHDR algorithm that incrementally builds a decision tree or regression tree, which serves as the robot's brain, to store its historical experience for very high-dimensional regression or decision spaces by an online real-time learning system. Subsequently, the validity of the autonomic mental development and IHDR algorithm was verified on the SAIL and DAV robot platforms of Michigan State University [16,17]. On the basis of the IHDR algorithm, Wu et al. [18] developed a system that builds an IHDR tree using landmark information, which is combined with its geographic position by a description vector of feature points to enable unmanned combat aerial vehicles (UCAV) to save and recall perception information. Aiming to address the shortcoming that traditional navigation robots are generally applied to static scenarios and have poor adaptability, Zhang et al. [19] proposed adaptive navigation methods under a dynamic scenario based on IHDR, which overcomes the over-dependence on the environment model in traditional navigation methods by incorporating its barrier strategy. The wearable affective robot Fitbot created by Chen et al. [20] can realize the EEG-based cognition of the user's behavior. As it acquires more data to enrich its model, Fitbot can recognize the user's intention and further understand the behavioral motivation behind the user's emotion. Chen at al. [21] proposed a task-driven development model to realize the autonomous perception of the environment with a physical layer, signal layer, and development layer. Ren at al. [22]

embedded intrinsic motivation into the autonomic development algorithm to improve initiatives in the robotic learning process, and two-wheeled robots were able to learn motion balance control by interacting with the environment. To date, the research on the autonomous development of robots has mainly focused on the fields of image classification and processing [15,23], as well as robot navigation and route planning [18,24,25]. Relatively few studies have explored the service cognition of robots.

On the basis of the current research status, we present an autonomous cognition and development system of robot service based on a composite framework that combines the Sarsa algorithm and IHDR algorithm, with the technical support of the TurtleBot platform and a smart home in the service robot laboratory in Shandong University. The framework of the system is shown in Figure 1. Using dispersed and modeled context data, we developed the Sarsa algorithm based on reinforcement learning to train robots that are learning the skills of providing appropriate services, which can customize the context to the user, in a specific discrete context through continuous trials. In the proposed approach, the results of reinforcement learning are stored in the q table. After that, an IHDR tree is built on the basis of the modified IHDR algorithm by mapping the discrete context information and service in the q table. Then, the context data provided by a large number of sensing devices in the intelligent space is used for service cognition, the expansion of the IHDR tree, and the autonomous cognitive development of service tasks. Finally, if the IHDR tree provides unsuitable cognition results, then the results are combined with user feedback. The combined cognition results are added to the IHDR tree to enrich the historical knowledge of the robots, thereby addressing the insufficiency of the IHDR knowledge base and realizing the further autonomous cognitive development of service skills in robots.

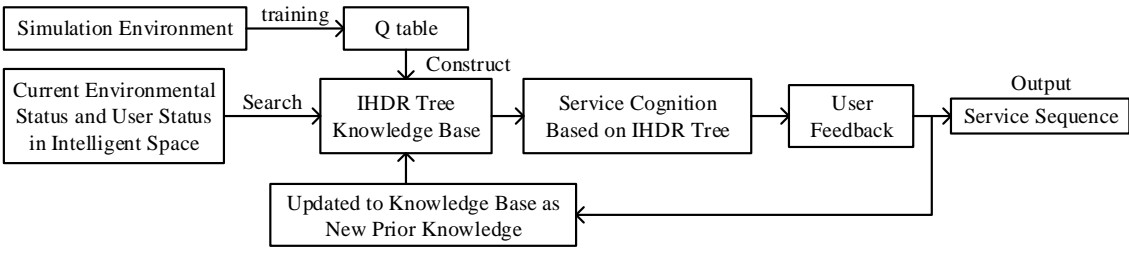

**Figure 1.** Learning and development of the service cognition framework for home service robots.

This paper is organized as follows. In Section 2, the process of robot service cognition training based on reinforcement learning is described. Section 3 describes another part of the service cognition system that is based on an autonomous development algorithm. Section 4 details two relevant experiments and analyzes the performance of the system. Finally, Section 5 concludes this paper.

## 2. Service Cognition Based on Reinforcement Learning

Robots in a new context are similar to newborn babies: their brains lack previous experience. Babies learn about their surroundings and gain experience by constantly interacting with an unfamiliar context and receiving rewards from it, and robots learn in the same way. When a robot does not have any experiential knowledge about the context, it explores and understands the context through a reinforcement learning algorithm. Because robots randomly select executable actions in the early stage of exploration, to avoid disturbing the user's activities, we model the context first to allow robots to explore the model with reinforcement learning. With the model, robots can learn how to provide appropriate services that change the status of the context so that it reaches the user's comfort level. This provides the robot with experiential knowledge for subsequent skill development in real space.

### 2.1. Model of Reinforcement Learning

Reinforcement learning is a kind of autonomous learning algorithm that allows the robot to select executable actions in the environment at random and receive rewards from the environment. By constantly trying and learning through rewards, the robot eventually recognizes the environment

and knows the actions that will change the environment to achieve its goals [26]. Figure 2 is a process diagram of reinforcement learning, and it shows that the reinforcement learning model requires four elements: the model of the context, the policy, and the reward and value functions.

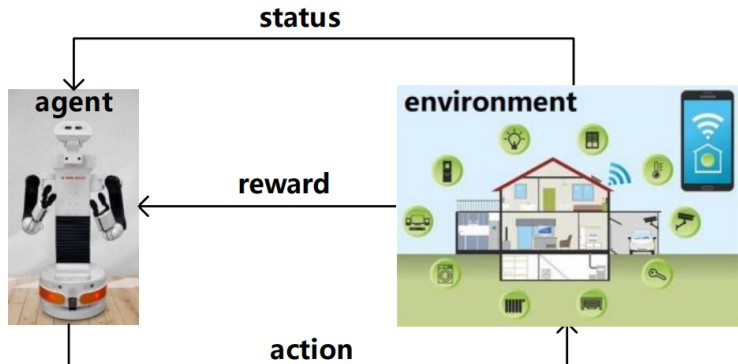

**Figure 2.** Reinforcement learning model.

### 2.1.1. Model of the Context

The context information in the modeled context includes information about (i) the natural environment, which can be received by various sensors; (ii) the user's status, which can be recognized by robots; and (iii) the open/closed or on/off status of devices or furniture, which can be returned by IoT. Since the natural environmental information is composed of continuous variables, it needs to be discretized, as shown in the discrete value column in Table 1. The user status and device switches are discrete variables and do not need extra processing. This information is shown in the coding column in Table 1. At present, the criteria for discretizing context information are formulated through 500 questionnaires. In the subsequent stage of self-development, adjustments are made according to actual user feedback.

### 2.1.2. Policy

In this article, actions are services that robots can provide, as shown in Table 2, in which the suffix "L" represents equipment located in the living room, the suffix "B" represents equipment located in the bedroom, "AC" is air conditioning and "TV" is television. For example, the "ACL" represents the air conditioning in the living room and the "TVB" represents the television in the bedroom. Robots should provide the appropriate service to change the status of the indoor environment and devices to make the context comfortable, and they should not implement any actions that bother the user and force the user to change his or her status. According to common sense, we assume that the context will change or remain the same after the robot performs a certain action. The context changes with the action of the robot, as shown in Table 3.

In the reinforcement learning process, the robot needs to achieve a balance between exploration and exploitation. Exploration means exploring the unknown scenario, which is conducive to gaining a full understanding of the scenario, but too much exploration leads to low learning efficiency. Exploitation means selecting the highest-reward action so far, which is conducive to improving learning efficiency, but it is possible to get the local optimal solution and miss the global optimal solution. The robot needs to choose the appropriate action selection policy to improve its learning efficiency as much as possible and avoid missing the global optimal solution.

The action selection policy used in this paper is as follows: If all the selectable actions have been tried in a scenario, then choose the action with the highest reward; otherwise, select the action with the highest reward with a probability of of $\varepsilon$ and randomly select the action that has not been tried with a probability of $(1 - \varepsilon)$, where $\varepsilon$ is given in $\sqrt{2/t}$ and decreases as the time of exploration increases.

**Table 1.** Discretization and coding of context information.

| Dimensionality | Discrete Value | Interval | Coding |
|---|---|---|---|
| Time | early morning | 7:00–8:00 | 1/0 |
| | morning | 8:00–11:30 | 2/0.2 |
| | noon | 11:30–13:30 | 3/0.4 |
| | afternoon | 13:30–18:00 | 4/0.6 |
| | evening | 18:00–21:00 | 5/0.8 |
| | night | 21:00–7:00 | 6/1 |
| Temperature | code | −10–25 °C | 1/0 |
| | suitable | 25–30 °C | 2/0.5 |
| | hot | 30–50 °C | 3/1 |
| Illumination | light | >550 Lx | 1/0 |
| | suitable | 100–550 Lx | 2/0.5 |
| | dark | 0–100 Lx | 3/1 |
| Location | table | - | 1/0 |
| | sofa | - | 2/0.33 |
| | bad | - | 3/0.67 |
| | dinningTable | - | 4/1 |
| Action | sit | - | 1/0 |
| | lie | - | 2/0.5 |
| | walk | - | 3/1 |
| behavior | read | - | 1/0 |
| | watchTV | - | 2/0.33 |
| | sleep | - | 3/0.67 |
| | eat | - | 4/1 |

**Table 2.** Executable action for robots.

| Sort | Service Name |
|---|---|
| Open XX | TurnOnLight, TurnOnLamp, TurnOnTVL, TurnOnTVB, OpenCurtain, OpenWindow, TurnOnACL, TurnOnACB |
| Interaction | SendDrinks, CallToWakeup |
| Close XX | TurnOffLamp, TurnOffTVL, TurnOffTVB, CloseWindow, CloseCurtain, TurnOffLight, TurnOffACL, TurnOffACB |

**Table 3.** The status of the context caused by the action.

| Service | Changes in the Context |
|---|---|
| TurnOnACL | Temperature in the living room becomes suitable |
| TurnOnLight | Indoor brightness becomes suitable |
| TurnOnTVB | TV status in bad room is on |
| CloseWindow | Window status is off |

In the early stage of reinforcement learning, the scenario is completely unknown to the robot, so it needs more exploration than exploitation. As the time of exploration increases, the robot develops some knowledge about the scenario, so it needs more exploitation and less exploration.

### 2.1.3. Reward and Value Functions

In this study, the goal of reinforcement learning is to provide services that make the scenario comfortable, and the equipment should have a different status at different times or different outdoor natural environmental parameters. Additionally, services that robots provide in a smart home can change the indoor scenario but not the outdoor scenario; therefore, this article enumerates the discrete outdoor natural environment and lists the equipment statuses that can make the indoor scenario

comfortable. In Table 4, the eight numbers in the equipment status column represent the switch status of windows, curtains, TVL, TVB, ACL, ACB, lamp, and ceiling light, respectively, where "1" is the on/open status, and "2" is the off/closed status.

**Table 4.** Appropriate hardware status in different scenarios.

| Environment State | | | User State | | | Hardware Status |
|---|---|---|---|---|---|---|
| Time | Temperature | Illumination | Location | Action | Behave | |
| 5 | 1 | 3 | 1 | 1 | 2 | 22212112 |
| 6 | 2 | 3 | 3 | 2 | 3 | 22222222 |
| 5 | 3 | 3 | 2 | 1 | 2 | 11222222 |

The reward is the key part of guiding robot learning, and the right reward strategies help to improve its learning efficiency. S defines the current status, and S_ is the last status: when the robot carries out an action and changes the state of the scenario from S_ to S, we return a reward according to this change, as well as the gap between states.

The gap between state S and the optimal state is defined as the distance, which is the number of actions that the robot needs to carry out to change the current scenario to the optimal scenario. Because of the uncontrollable status of the external natural environment, we choose the optimal state whose context information is the same as the current context information as the optimal for comparison.

For example, if S is at night, the temperature is suitable, the light is off, and the user is in bed, lying down, and sleeping, then the hardware status is "21221222". In the second line in Table 4, for status S, since the status of all hardware should be off, the robot should turn off or close the two "open" equipment switches. Hence, the distance between S and the optimal state is 2. The reward is defined as follows:

1.  If S = S_ or distance(S) = distance(S_) after the robot executes a service, then the robot executed an invalid service, so reward_ss = −10. For example, if the natural outdoor temperature is appropriate and the TurnOnACL service is carried out, then the scenario status does not improve.
2.  If distance(S) > distance(S_) after the robot executes a service, then the current scenario is worse than the optimal context, indicating that the wrong service was executed, so reward_ss = −10. For example, if the robot turns off the originally opened air conditioner when the natural outdoor temperature is unsuitable, then the status of the scenario becomes worse. On the contrary, if distance(S) < distance(S_) after the robot executes a service, then reward_ss = +10.
3.  If S is exactly the optimal status after the robot executes a service, then the current episode ends, and reward_rs = 10. For example, if the scenario is the same as the second line in the table, except the status of the window is "1", then the robot executed the OpenWindow service, making the window status "2".
4.  If the robot executes an action that is explicitly prohibited, then reward_sa = −10. An example of a prohibited action is opening a curtain at night.

The total reward can be obtained by the weighted summation of the above three rewards as follows.

$$reward\_total = 0.5 * reward\_rs + 0.2 * reward\_ss + 0.3 * reward\_sa. \tag{1}$$

*2.2. The Q Table*

The input space is encoded to record the scenario status in the q table. In contrast to the 0–1 coding method [4], the coding mode of the following table maintains the original dimensionality of the data to avoid increasing the dimensionality of the q table. At the same time, the coded data should be normalized by the 0–1 normalization method. The processed results are shown in Table 1. In the coding column, the number to the left of "/" is the value before normalization, and the number on the right is the normalized value.

With this coding method, the index line number of each scenario status in the q table is in accordance with Formula (2). The 18 columns in the q table correspond to 18 different services, as shown in Table 2:

$$index = \sum_{i=0}^{n-1} (num[i] - 1) * 2^i. \tag{2}$$

### 2.3. The Sarsa Algorithm

Since the scenario status is a discrete variable, we chose to use the Sarsa algorithm to explore the scenario. Algorithm 1 is the pseudo-code of the Sarsa algorithm, and the q table is saved locally after the Sarsa training is completed.

---

**Algorithm 1:** Sarsa Algorithm

---

Initialize Q(s,a) arbitrarily
Repeat (for each episode):
    Initialize s
    Repeat (for each step of episode):
        Choose a from s using policy derived from Q(e.g., e-greddy)
        Take action a, observe r, s$'$
        Choose a$'$ from s$'$ using policy derived from Q(e.g., e-greddy)
        Q(s,a) $\leftarrow$ Q(s,a) + $\alpha$[r + $\gamma$ Q(s$'$,a$'$) - Q(s,a)]
        s $\leftarrow$ s$'$, a $\leftarrow$ a$'$
        until s is terminal

---

## 3. Service Cognition Based on the Autonomous Development Algorithm

All the scenario statuses in the q table are traversed, the service list that needs to be executed to change the current scenario to the optimal scenario is acquired through the Sarsa algorithm, and the mapping between the scenario and services is obtained as sample data. From the sample data, the IHDR tree is constructed as a knowledge base for storing the robot's historical experience. Then, the robot recognizes the service by analyzing the sample data and the historical experience in the IHDR tree. The process includes three main steps: collecting and processing sample data, constructing the IHDR tree, and searching the IHDR tree. These sample data have the characteristics of continuous and discrete mixing distributions. However, the traditional IHDR algorithm is only suitable for continuous data, so we modified the traditional IHDR algorithm to solve this problem.

### 3.1. Collecting and Processing Sample Data

Developing robots that learn independently through reinforcement learning rather than the user's guidance effectively reduces the frequency of human–computer interaction and avoids disturbing the user's life.

The autonomous cognition of robotic service tasks can be seen as a mapping from the current scenario information and the user's state to the services that the user requires. Therefore, the sample data can be divided into two parts: the input space

$$X = (EnvironmentState, UserState, HardwareState) \tag{3}$$

and the output space

$$Y = (ServiceUsereeded). \tag{4}$$

The specific features contained in $X$ and $Y$ are shown in Tables 1 and 2; the input space of the sample data are 14-dimensional, and the output space is 18-dimensional. It is time-consuming for robots to understand such high-dimensional data and map the input space to the output space, which leads to a decrease in the learning and development efficiency. Therefore, the sample data are

processed according to custom rules and stored in a MySQL database following the principle of "only concentrate on the necessary conditions for a service instead of other unrelated conditions". Some of the processed data are shown in Table 5, where "0" means that this feature is an unrelated condition for its corresponding service.

**Table 5.** Processed sample data.

| Environment State | | | User State | | | Hardware Status | Service |
|---|---|---|---|---|---|---|---|
| Time | Temperature | Illumination | Location | Action | Behave | | |
| 0 | 0 | 170 | 0 | 0 | 0.33 | 00000002 | TurnOnLight |
| 0 | 0 | 0 | 0 | 0 | 0 | 00000020 | TurnOnLamp |
| 7 | 0 | 0 | 0 | 0 | 0 | 02000000 | OpenCurtain |
| 22 | 0 | 0 | 0 | 0 | 0 | 10000000 | CloseWindow |
| 0 | 32 | 0 | 0 | 0 | 0 | 10000000 | CloseWindow |

### 3.2. Analysis of the Modified IHDR Algorithm

We rely on the mapping

$$f : Y \to X \tag{5}$$

shown in Figure 3, in which the numbers in the input space $X$ represent different scenarios. Only four maps are selected in Figure 3 for illustration. The IHDR algorithm can cluster the output space and input space simultaneously to realize the classification and regression of high-dimensional data. In particular, during the IHDR tree construction process, we can cluster the input space $X$ (current scenario) according to the same the output space $Y$ (service) to achieve regression; during the IHDR tree searching process, we can find the cluster to which the input space $X$ (current scenario) belongs, and the output of this cluster is the output of $X$, which represents the services that the robot should provide in the current scenario, thereby realizing classification. The definition of the input and output is

$$X = (Time, Temperature, Illumination, Location, Action, Behavior, HardwareState),$$
$$Y = (ServicePriority, ServiceName). \tag{6}$$

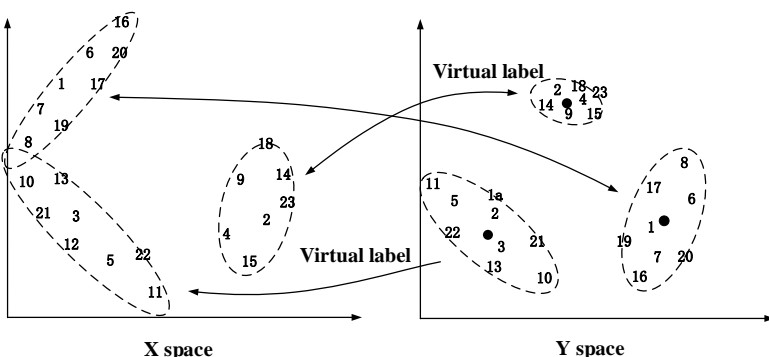

**Figure 3.** Mapping $f : Y \to X$.

As shown in Figure 4, the IHDR algorithm is divided into three main parts: storing the tree, updating the tree, and searching the tree. The modification of the IHDR algorithm presented in this paper is mainly reflected by the ability to process discrete data; the details are specified in Step 3 of Algorithm 2 and Step 3 of Algorithm 3.

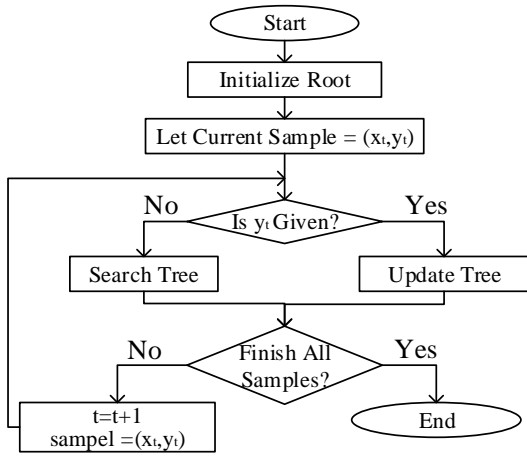

**Figure 4.** Flow chart of the IHDR algorithm.

### 3.2.1. Store the IHDR Tree

In our approach, a MySQL database is used to store the IHDR tree. In the traditional method of storing a tree in a database, at least one column of features is added to the sample to show the node's depth information in the tree, which increases the dimensionality of the data, stretches the SQL statement, and makes the database difficult to maintain and manage. In order to avoid the drawbacks mentioned above, we store the IHDR tree in a new database; the nodes are stored in the tables of the database, and the nodes' depth information is indicated by the table names. For example, "$Y_{nm}$" is the mth child node of the nth node of the first layer, named "$Y_n$". The structure of the tree is shown in Figure 5.

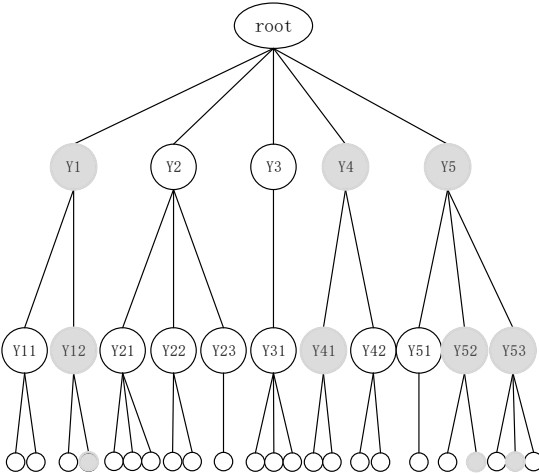

**Figure 5.** Searching process of the IHDR algorithm.

### 3.2.2. Update IHDR Tree

Compared with the traditional batch learning method, the IHDR algorithm builds the tree incrementally using an incremental learning method. The data with the same output will be clustered to the same input, and robots' knowledge base is constructed incrementally. The knowledge base is also called an IHDR tree as it stores the knowledge as a tree structure. The modified process of updating an IHDR tree is mainly reflected in adding the ability to process discrete data. For the unrelated feature, set it equal to 0 directly; for the necessary feature, if it is continuous data, we update her mean and range as usual; otherwise, update the list recording all possible discrete values in this feature. See Step 3 in Algorithm 2 for details.

---

**Algorithm 2:** Update Tree Algorithm.

---

**Input:** A set of sample data S with input and output;
**Output:** Cluster table in MySQL database and .csv files;
**Step 1:** Initialization;
    **Step 1.1:** Initialize root;
    **Step 1.2:** Let A be the active node waiting to be searched, and A is initialized to be the root;
    **Step 1.3:** Traverse samples and let S=(Xt,Yt) be the current sample;
**Step 2:** Update tree;
    **Step 2.1:** Traverse all the child nodes of A, find the out service type node (internal node) or service node (leaf node) to which Yt belongs, and let it be the new A;
    **Step 2.2:** While A is an internal node, switch to step 3, and switch to step 2 after doing step 3;
    **Step 2.3:** While A is a leaf node, switch to step 3;
**Step 3:** Update cluster. The clustering information is updated by Xt, which is convenient for classifying samples when searching the tree. The cluster information is stored in a .csv file with the same name as the node;
    **Step 3.1:** Traverse all the features of Xt, and set the value of the current feature equal to T;
    **Step 3.2:** If the T's corresponding bit in the .csv file is 0, continue;
    **Step 3.3:** If T=0, then T is the unrelated condition of the corresponding service; set T's corresponding bit in the .csv file equal to 0, continue;
    **Step 3.4:** If T is continuous, update its mean and range;
    **Step 3.5:** If T is discrete, update the list of all possible values of T.

---

Partial clustering information of some nodes is shown in Table 6.

**Table 6.** Partial clustering information of nodes.

| Info | WindowState | UserLocation | Temperature | Illumination |
|------|-------------|--------------|-------------|--------------|
| $Y_1$ | 0 | [bad, sofa] | 0 | 0 |
| $Y_{21}$ | 0 | 0 | 0 | 108.4, 91.6 |
| $Y_{25}$ | 2 | 0 | 20.50, 5.5 | 0 |

The process of clustering is shown in Figure 6 [15]. Each point in the graph is a sample, and each circle is a cluster. There are two clusters in the graph, and all samples in the same cluster have the same output.

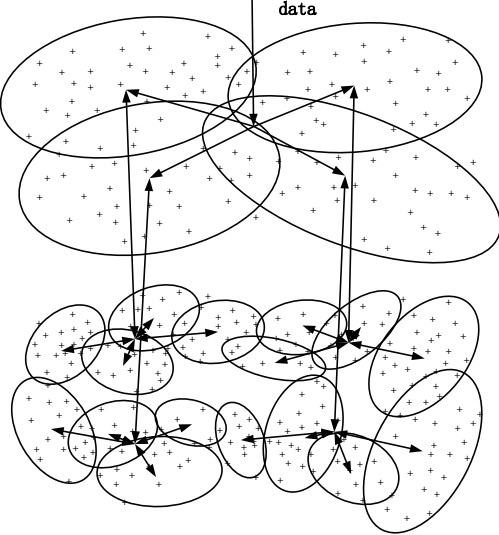

**Figure 6.** Clustering process of the IHDR algorithm.

### 3.2.3. Search the IHDR Tree

When $Y_t$ in the sample S = ($X_t$, $Y_t$) is unknown, it can be obtained by searching the IHDR tree to find the leaf node that matches $X_t$; the output of this leaf node is $Y_t$. The modified search process of the IHDR tree is shown in Algorithm 3. The process of retrieving the tree is shown in Figure 5, in which the shaded nodes are true nodes, and the searching process is as follows. Let all nodes in the first layer be the active nodes set, and the true nodes set, which is $Y_1$, $Y_4$, $Y_5$, can be obtained by computing the response of $X_t$. Then, let all children nodes of $Y_1$, $Y_4$, and $Y_5$ be the new active node set; the response is thus similarly computed. The rest of the process can be carried out in the same way. Finally, the three shaded leaf nodes form the set of the last true nodes, whose output can be regarded as the output of $X_t$.

---

**Algorithm 3:** Update Tree Algorithm.

---

**Input:** A set of sample data S without output;
**Output:** A service sequence SS;
**Step 1:** Initialization;
    **Step 1.1:** Define an active node set as a set of active nodes waiting to be searched, and the active node set is initialized as the set of all nodes in the first layer of the IHDR tree;
**Step 2:** Search tree;
    **Step 2.1:** Traverse the active node set, find the nodes that respond to (Xt as true by calling step 3, and let these nodes be the true node set;
    **Step 2.2:** If all nodes in the true node set are leaf nodes, then SS is the output set of all these leaf nodes, and the searching process of the current sample ends;
    **Step 2.3:** Else, clear the current active node set, add all leaf nodes and all child nodes of internal nodes in the true node set to the active node set, and repeat step 2.
**Step 3:** Calculate the response of ($X_t$ in sample S to some node's cluster information. The result "true" means that S has exactly the same input as this node, and S's output should be the same as this node's output;
    **Step 3.1:** Traverse all the features of (Xt and set the value of the current feature equal to T;
    **Step 3.2:** The feature value in the cluster information corresponding to T is "0", continue;
    **Step 3.3:** If the feature value in the cluster information corresponding to T is a list, and T belongs to the list, then continue. Else, return false;
    **Step 3.4:** If the feature value is mean and range, and if T satisfies Formula 7, continue. Else, return false;
    **Step 3.5:** Return true if the traversing is over.

---

The modification of the algorithm also follows the principle of "only concentrate on the necessary conditions for a service instead of other unrelated conditions", so we are not concerned with features whose value is 0. A continuous feature should have a value that is in the same range as one cluster; a discrete feature should have a value that belongs to the list that records the possible values. If it belongs to this cluster, then its output is the same as the cluster's output. Step 3 in Algorithm 3 shows these details:

$$T \in [ave - range, ave + range]. \tag{7}$$

Thus, the IHDR algorithm is complete. As the robot continues to process increasingly more sample data, the tree is incrementally constructed and expanded. The more "flourished" the IHDR tree, the richer the historical experiential knowledge stored in the tree, and the greater the service recognition of the robots. By analyzing the theory of the IHDR algorithm, we can conclude that it updates its historical experience using an incremental learning method, which is different from the traditional batch learning method, such as that used in a neural network. If the robot must learn a lot of new sample data in the home environment, then it only needs to learn the added data and store new experience in its human-brain-like knowledge base using this incremental learning method rather

than learn all of the sample data again using the batch learning method. This is more in line with the process of human learning and mental development.

### 3.3. Service Plan Based on the Hierarchical Task Network

By decomposing complex tasks layer-by-layer, the hierarchical task network (HTN) can output a series of atomic tasks that can be executed by the robot directly. JSHOP2 [27] is a planner that can realize the HTN. The JSHOP2 planner is composed of a domain description file and a problem description file. The domain description file defines the method of task decomposition by a keyword "method", and the changes in state information occur during atomic task execution by a keyword "operator". In the problem description file, the current scenario information and the complex tasks are described. Figure 7 provide details using the "SendDrinks" example to show the process of task decomposition in JSHOP2.

```
(def domain basic (
   (:method (SendDrinks ?p)
      ( (in ?p ?person_pos) (in ?cup ?cup_pos)
        (in ?cup_pos ?cup_pos_pos) (havewater ?water_dispenser)
        (in ?water_dispenser ?dispenser_pos) )
      ( (gopick ?cup ?cup_pos) (handto ?p ?person_pos ?cup)
        (pourwater ?water_dispenser ?cup) ) )
   (:method (gopick ?object ?obj_pos)
      ( (in ?object ?obj_pos) (in ?obj_pos ?obj_pos_pos))
      ( (!moveto ?obj_pos ?obj_pos_pos) (!pick ?object ?obj_pos) ) )
   ……
   (:operator (!moveto ?object ?obj_pos)
      ( (in ?object ?obj_pos))
      ()
      ())
   (:operator (!pick ?object ?obj_pos)
      ( (in ?object ?obj_pos))
      ( (in ?object ?obj_pos))      ;add this state
      ( (have ?object)) )           ;delete this state
))
……
```

**Figure 7.** Domain description file.

### 3.4. Provide User Feedback and Update the IHDR Tree Online

The lack of experience during service recognition by the IHDR tree is addressed by adding user feedback, thereby allowing the user to correct the inaccurate service.

A graphical human–machine interface is used for the user to input the real command if the robot provides unreasonable services. When the user is not satisfied with the current service sequence, the user expresses what he or she really wants through the interface, and the correct service is added to the service sequence and incrementally added to the IHDR tree after the data have been processed. Combined with the user's guidance, the service sequence is regarded as a new learning experience of the robot and referenced later; this process is also important for the cognitive development of the robot's service skills.

In the final part of this study, the validity and practicability of the system were verified on the TIAGO robot using the gazebo simulation software (Gazebo 9.0, the University of Southern California, Los Angeles, State of California, US) and TurtleBot robot in the Service Robots Laboratory of Shandong University.

## 4. Simulation Experiments and Results Analysis

In our method, the simulation context model is first established, and the Sarsa algorithm based on reinforcement learning is then used to train the robot to provide services, with the obtained sample data used as the guidance data for the early development of IHDR trees. Then, the current environmental state, the user state, and the hardware state are collected by the sensor and the IoT in the smart home as the input space, and the service is recognized by the historical experience stored in the IHDR tree as the output space. Both the input and output data are used to construct the IHDR tree. The IHDR tree is stored in a MySQL database, which is visualized by Navicat for MySQL's powerful graphical interface. The database is updated by using Python's MySQLdb interface. The IHDR tree is updated and searched by Python3, which is combined with the user's feedback to realize the cognition and development of the robot's service task automatically. The simulation experiment was based on the TIAGO robot using the gazebo simulation software, and the algorithm experiments were verified based on the TurtleBot robot.

### 4.1. Results of Service Cognition Based on the Sarsa Algorithm

The q table is initialized as an all-zero matrix, the agent is trained according to the reinforcement learning model in Chapter II, and the learning process is printed to the interface. In the early stage of training, the agent has no experience, so in a random initial context, it needs to go through multiple episodes to achieve the optimal context, as shown in Figure 8. After training for a period of time, the agent has gained some experience and needs fewer episodes to achieve the optimal context, but it still makes a few wrong decisions and receives negative rewards. After the agent has been training for a long time, the service decision can be made accurately, as shown in Figure 8.

```
training 0: initial scenario: 112-323-2222-2222
    environment: 112-323-2222-2222, action: TurnOnLight, new_environment: 112-323-2222-2221, reward: -2
    environment: 112-323-2222-2221, action: TurnOnLight, new_environment: 112-323-2222-2221, reward: -2
    environment: 112-323-2222-2221, action: TurnOnTVL, new_environment: 112-323-2212-2221, reward: -2
    environment: 112-323-2212-2221, action: TurnOnTVL, new_environment: 112-323-2212-2221, reward: -2
    environment: 112-323-2212-2221, action: CloseWindow, new_environment: 112-323-2212-2221, reward: -2
    environment: 112-323-2212-2221, action: OpenCurtain, new_environment: 112-323-2112-2221, reward: 2
    environment: 112-323-2112-2221, action: TurnOnLight, new_environment: 112-323-2112-2221, reward: -2
    environment: 112-323-2112-2221, action: TurnOnLight, new_environment: 112-323-2112-2221, reward: -2
    environment: 112-323-2112-2221, action: TurnOffACB, new_environment: 112-323-2112-2221, reward: -2
    environment: 112-323-2112-2221, action: OpenCurtain, new_environment: 112-323-2112-2221, reward: -2
    environment: 112-323-2112-2221, action: OpenWindow, new_environment: 112-323-1112-2221, reward: -2
......
training 300: initial scenario: 523-111-1122-2222
    environment: 523-111-1122-2222, action: TurnOnLight, new_environment: 523-111-1122-2221, reward: 2
    environment: 523-111-1122-2221, action: CloseWindow, new_environment: 523-111-2122-2221, reward: -2
    environment: 523-111-2122-2221, action: TurnOnACL, new_environment: 523-111-2122-1221, reward: -2
    environment: 523-111-2122-1221, action: TurnOnLamp, new_environment: 523-111-2122-1211, reward: 2
    environment: 523-111-2122-1221, action: OpenWindow, new_environment: 523-111-1122-1211, reward: 2
    environment: 523-111-2122-1221, action: TurnOffACL, new_environment: 523-111-1122-2211, reward: 7
......
training 9000: initial scenario: 231-212-2112-2222
    environment: 231-212-2112-2222, action: TurnOnACL, new_environment: 231-212-2112-1222, reward: 2
    environment: 231-212-2112-1222, action: OpenWindow, new_environment: 231-212-1112-1222, reward: 7
```

**Figure 8.** Reinforcement learning training process.

### 4.2. Results of Service Cognition Based on Autonomous Development Algorithm

By traversing all rows in the q table, the discrete contextual variables are mapped to randomly selected values with the corresponding range as the input, and the services that should be executed to

change the scenario from the current status to the optimal status are the output. The sample data are constructed as in Table 7.

    The IHDR tree is built by using the sample data in Tables 7 and 8. There are two-layer nodes in this IHDR tree stored in the MySQL database, and there are five internal nodes in the first layer that represent five categories of service, $Y_n$_cluster (n = 1, 2, 3, 4, 5). There are 23 internal nodes named $Y_{nm}$_cluster in the second layer, representing 23 different services. The cluster information of each node is stored in a .csv file whose number is the same as the node's number.

    After completing the construction of the IHDR tree, four moments are selected randomly to collect information about the actual scenario in the smart home, which is based on recognizing the service autonomously by referring to historical knowledge stored in the IHDR tree. The simulation results are shown in Figure 9, and the three states of the data are described below. The actual scenario information is as follows:

1.  Contextual state: the current time is 07:00 a.m., the outdoor temperature is 25 °C, and the illumination intensity is 100 Lx;
2.  User state: the user's location is "bad", the user's action is "lie", and the user's behavior is "sleep";
3.  Hardware state: the states of all hardware are "off", including the window, curtain, TVL, TVB, ACL, ACB, desk lamp, and top light.

    After searching the IHDR tree, the robot provides a sequence of prioritized services, which are waking the user up and opening the curtains and windows.

```
Run    runme
►  ↑    D:\ananconda\install\python.exe E:/Python/IHDR_hm/runme.py
■  ↓    scenario information:
         ( 1 , 22 , 19 , 200 , chair , sit , read , 1 , 1 , 2 , 2 , 2 , 2 , 1 , 1 )
        service sequence cognized:
         ['SendDrinks', 'RemindToSleep', 'CloseWindow', 'CloseCurtain']

        scenario information:
         ( 2 , 19 , 30 , 200 , sofa , sit , watchTV , 1 , 1 , 2 , 2 , 2 , 2 , 2 , 2 )
        service sequence cognized:
         ['TurnOnLight', 'TurnOnTVL', 'TurnOnACL', 'SendDrinks', 'CloseWindow']

        scenario information:
         ( 3 , 7 , 25 , 100 , bad , lie , sleep , 2 , 2 , 2 , 2 , 2 , 2 , 2 , 2 )
        service sequence cognized:
         ['CallToWakeup', 'OpenCurtain', 'OpenWindow', 'SendDrinks']

        scenario information:
         ( 4 , 11 , 22 , 300 , dinningTable , sit , eat , 1 , 1 , 1 , 2 , 2 , 2 , 2 , 2 )
        service sequence cognized:
         ['SendDrinks', 'TurnOffTVL']
```

**Figure 9.** Service cognition results using the IHDR tree.

**Table 7.** Normalized input data.

| Environment State | | | User State | | | Hardware Status |
|---|---|---|---|---|---|---|
| Time | Temperature | Illumination | Location | Action | Behave | |
| 0.78 | 0.78 | 0.1 | 0.25 | 0 | 0.25 | 1122 2222 |
| 0.26 | 0.68 | 0.6 | 0.5 | 0.5 | 0.5 | 2222 2222 |
| 0.43 | 0.81 | 0.6 | 0 | 0 | 0 | 1122 2222 |
| 0.91 | 0.49 | 0 | 0.5 | 0.5 | 0.5 | 1121 2222 |
| 0.35 | 0.59 | 0.6 | 1 | 0.5 | 1 | 1122 2222 |

**Table 8.** Output sample data.

| Open XX | Interation | Close XX |
|---|---|---|
| 1010 0010 | 10 | 0001 0000 |
| 0000 1100 | 10 | 0000 0000 |
| 0100 0001 | 10 | 0001 0000 |
| 0000 0000 | 00 | 0011 1000 |
| 0000 0000 | 00 | 0000 0000 |

### 4.3. Performance Analysis of the Service Cognition and Development System

In this study, we tested the time and quality of the system's service cognition and compared them with those of the Conflict-driven Hierarchical Meta-CSP Planner (CHIMP) [28] and the personalized recommendation method based on the domain knowledge map [29]. The experimental data in [28] were estimated by GetData Graph Digitizer software (2.26).

As shown in Figure 10, when the number of operators is less than three, our proposed service cognition method takes slightly more time than the CHIMP method. When the number of operators is more than four, our proposed method has certain advantages in time.

The quality of service cognition is represented by the precision, recall, and F value. We compared the services that the user really wants with the service provided by the robots and calculated the precision, recall, and F value of the Sarsa+IHDR method. Then, the values were compared with those from the Kmeans method [29]. The results are shown in Table 9.

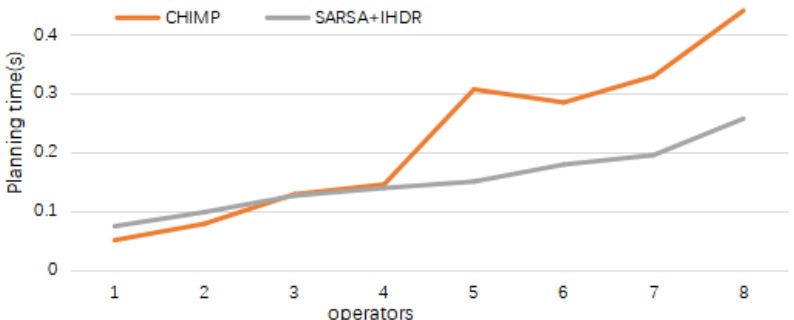

**Figure 10.** Comparison of service cognition and combination methods.

**Table 9.** Quality of the generated service.

|  | Precision (%) | Recall (%) | F (%) |
|---|---|---|---|
| Kmeans | 88.66 | 84.57 | 86.57 |
| Sarsa + IHDR | 98.21 | 90.26 | 94.07 |

It can be seen that the precision and recall rate of the Sarsa+IHDR method are both higher than those of the Kmeans method. With the gradual learning and development of the IHDR tree, the precision and recall rate will be improved, which is also of significance to autonomous development.

### 4.4. Service Cognition Based on the TurtleBot Robot

The autonomous cognition and development system was validated by applying it to the TurtleBot robot, which is shown in Figure 11 in the Service Robots Laboratory of Shandong University, by simulating the actual home environment of a smart home and IoT in the laboratory. Because of the limitations of the robot's hardware, the robot cannot provide services in practice; therefore, the following experiments are described with the assumption that the robot's arrival at the site where the service is performed is equivalent to its successful performance of the service.

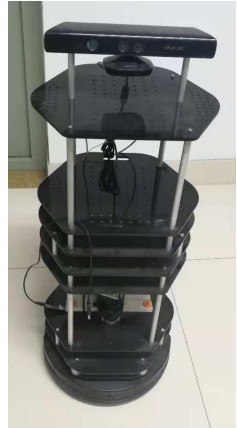

**Figure 11.** TurtleBot robot.

After logging onto the system, the human–machine interface of the autonomous cognitive system for service tasks shown in Figure 12 can be seen by the user. The interface is composed of three main modules:

1. "Current information of smart home" module: displays the current context information of the smart home (including the environmental state, user's state, and hardware state), which is returned by the sensing device and IoT in real time.
2. "Current services provided by robots" module: displays the autonomous service cognition result of the robot based on the IHDR tree.
3. "User-directed services" module: the user's guidance is given to the robot through this module when the services provided do not meet the needs of the user. The services that the user wanted can be selected from a drop-down menu, and the user-directed service combined with the robot-provided services can be added to the IHDR tree by clicking the "submit" button.

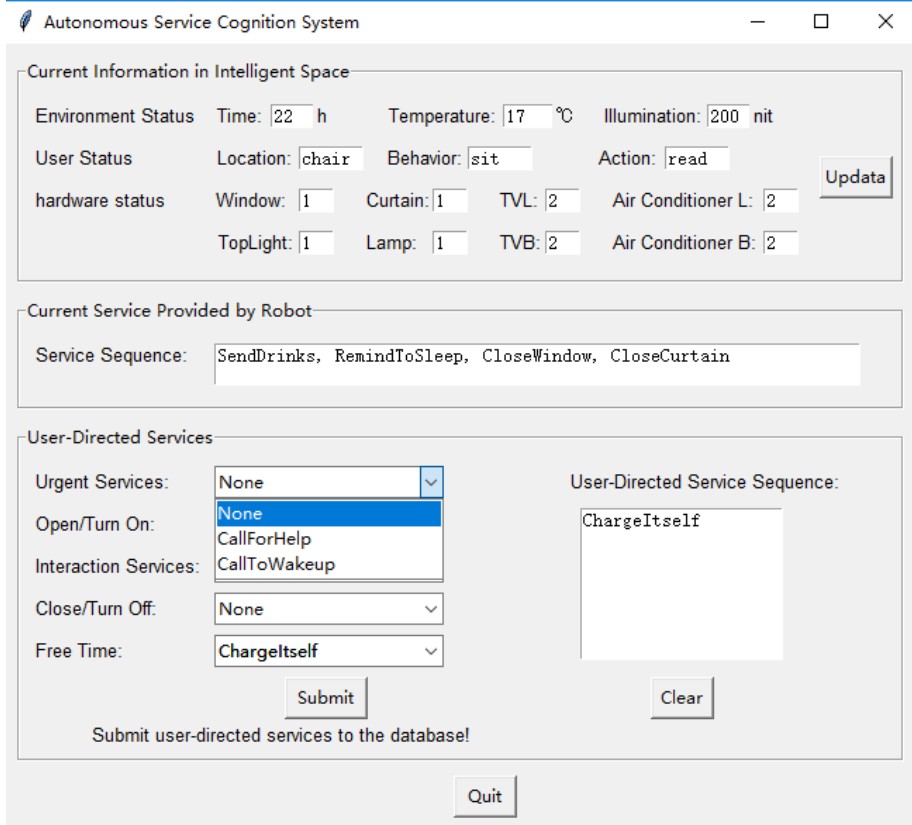

**Figure 12.** The human–machine interface of the autonomous cognition system for service tasks.

After the current user-directed services and the robot-provided services are combined, as shown in Figure 13, the service sequence is decomposed into a series of atomic services based on JSHOP2. The process and results are shown in Figure 13. Figure 13a shows the five initial complex tasks of the robot, Figure 13b shows the current state in the Current State column, and Figure 13c shows the final result of the decomposition. The indentation represents the level of task decomposition, and the atomic services preceded by a number are executed in order.

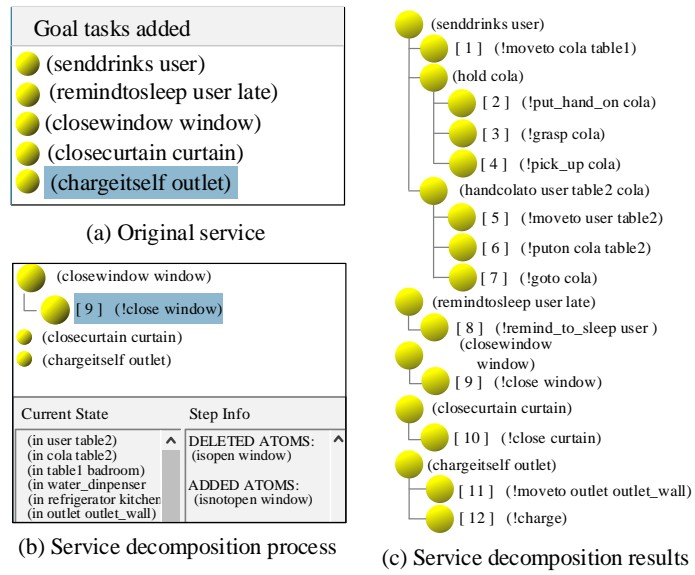

Figure 13. Decomposition of a task sequence based on JSHOP2.

According to the generated atomic service sequence, the simulation experiment based on robot operating system (ROS) is shown in Figure 14, which displays the map construction process of the TIAGO robot in the simulation context and the execution of the atomic service sequence based on the built-in navigation algorithm.

The trajectory of the robot as it completes the service tasks is shown in Figure 15. From the starting point, the robot performs the "SendDrink" service first: it goes to the water dispenser, waits for one minute (which means that it takes a glass of water), goes to the user, and provides the user with the water. It performs the "RemindToSleep" service. Then, the "CloseWindow" and "CloseCurtain" services are performed using IoT technology. Finally, it goes to the position of the socket to perform the "ChargeItSelf" service following the user's guidance.

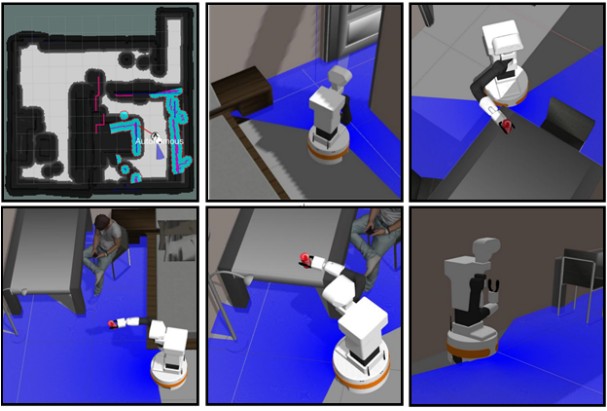

Figure 14. Simulation results based on ROS.

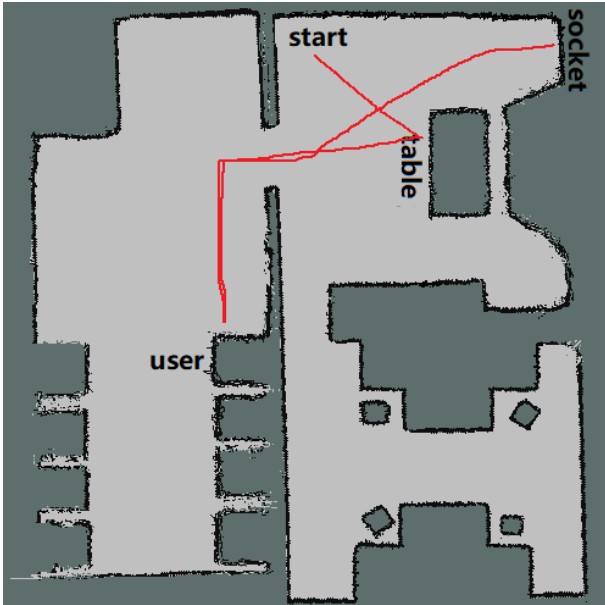

**Figure 15.** Path of the robot performing services.

## 5. Conclusions

In this paper, we proposed an autonomous cognition and development system of service robots based on a composite framework that combined the Sarsa algorithm and the IHDR algorithm. First, we took advantage of a feature of reinforcement learning, zero-sample learning, to build a context–action model of the home environment to enable the robot to improve service cognition skills in a discrete simulation context through constant exploration of the context using the Sarsa algorithm. The learning results were stored in the q table as context–action mapping pairs, which were regarded as the robot's experience, and these mapping pairs were appended to the IHDR tree incrementally as the robot's initial experience. Then, we put the robot in a real context to execute service cognition from its existing experience, and the current context information is offered by the IoT in a smart home. If there was experience in the IHDR tree similar to the current scenario, the robot recalled the service directly from its experience; otherwise, it accepted the user's guidance through the human–machine interface and added the current context and user-guided service mapping pair to the IHDR tree as a new experience. As the robot dealt with increasingly more scenarios, the IHDR tree was expanded incrementally and stored increasingly more of the robot's historical experience, and the robot's service cognitive ability developed gradually. However, through the human–computer interaction mentioned earlier, our implementation method requires the user to walk to the computer equipped with the robot for operation. Incorporating flexibility and adaptability into the human–computer interaction is the next improvement that will build on this work. Moreover, the personality of service robots for different users is not taken into account in this system at present. This feature can be realized in the near future to provide personalized services to different users.

**Author Contributions:** Conceptualization, F.L. and G.T.; methodology, X.L. and H.W.; software, M.H. and X.L.; validation, M.H. and W.S.; formal analysis, F.L. and X.L.; investigation, F.L. and M.H.; resources, G.T. and F.L.; data curation, M.H. and X.L.; writing—original draft preparation, M.H.; writing—review and editing, F.L. and W.S.; visualization, M.H. and H.W.; supervision, M.H. and W.S.; project administration, F.L.; funding acquisition, G.T. and F.L. All authors have read and agreed to the published version of the manuscript.

**Funding:** This research was funded by the National Natural Science Foundation of China (61973187, 61773239, 61973192), and the Taishan Scholars Program of Shandong Province.

**Conflicts of Interest:** The authors declare no conflicts of interest.

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
