# Peer review of "Learning and Development of Home Service Robots’ Service Cognition Based on a Learning Mechanism"

_applsci, doi:10.3390/app10020464_

Round 1

Reviewer 1 Report

Learning and Development of Home Service Robot’s

Service Cognition Based on Learning Mechanism

Review

The article suggest a model to develop  a home  service robot, that would offer some flexible services, adapted to a particular home environment marked by changes and incertitude. The approach lays in the area of the Internet of Things(IoT) technologies in the smart home and deals with training robots to over various services  that could adapt the user’s needs. Current work suggests a reinforcement learning algorithm based on the robot’s experience, which enhance the adaptability to a changeable environment. The logic of the training in this case is one in which the user assist the robot to get the necessary experience and learn from the past, by adjusting the improper services and enriching the robot experience with various  dynamics. The model is  based on a parallel with human cognition, in which by  reinforcing positive-outcome behaviors, people learn to  create  adaptive metal shortcuts.

The suggested model and approach are original and intriguing. Still  the article would benefit from some improvements

The writing style  is rather difficult to follow and lacks the English proof The article is rather technical and therefore the clarity  of text is important, also the easiness in reading it. The article has a relatively short conclusion at the end and does not discuss the implications of the current results A discussion section would be benefic at the end of the article, allowing the possibility to discuss and critically analyze the main findings Generally speaking a more critical approach is needed in the text. There is a parallel between human cognition and learning  process at robots and this need to be critically presented Also the user interaction in the training of the robot to allow  flexibility and adaptability need to be addressed Internet of Things(IoT) technologies is an aspect largely discussed in the literature. The reader expects more relevant literature on this direction

Reviewer 2 Report

The paper is interesting but seriously flawed by several typos, minors and poorly readable statements (see below). A deep proof-reading is therefore mandatory before a full evaluation of the proposal. Currently, the readability of the paper is my main concern and I would like to review it after the overall English would be smoothed. There are also several secondary concerns.
Relationships between the proposal and the related work reported in Introduction should be underlined. Generalize the reinforcement learning concept since currently it seems being related just to robot "Reinforcement learning is a kind of autonomous learning algorithm, making robot select the executable action randomly in the environment and get the reward from the environment. By constantly trying and learning from reward, the robot finally recognizes the environment and knows how to take actions to change the environment to achieve its goals....As the robot dealing with more and more scenarios, the IHDR tree is expanded incrementally and stores more and more robot’s historical experience, and the ability of the robot service cognition is developed gradually". As general reference for reinforcement learning cite <Mohammadi, Mehdi, et al. "Semisupervised deep reinforcement learning in support of IoT and smart city services." IEEE Internet of Things Journal 5.2 (2017): 624-635> and <Savaglio, Claudio, et al. "Lightweight Reinforcement Learning for Energy Efficient Communications in Wireless Sensor Networks." IEEE Access 7 (2019): 29355-29364>. Regulate the usage of acronym: once introduced, they should be always used (e.g., IoT acronym introduced in the Introudcion but "Internet of Things" appears in the conclusion). Indent code of algorithm 2 for a better graphical presentation. Cite <Ashraf, QAZI MAMOON, and MOHAMED HADI Habaebi. "Introducing autonomy in internet of things." 14th International Conference on Applied Computer and Applied Computational Science (ACACOS'15). 2015> and <Savaglio, Claudio, and Giancarlo Fortino. "Autonomic and cognitive architectures for the Internet of Things." International Conference on Internet and Distributed Computing Systems. Springer, Cham, 2015> when introducing autonomicity and cognitivity for the IoT. Use past tense verbs for the conclusion "In this paper we propose"->"In this paper we proposed". Fig 10, title and caption are the same: remove the title. Fig.15, instead, is poorly readable due to low quality. Other comments:
*missing space in "regression(IHDR)" and in "module.Displaying"
*edit "Robot can be truly intelligent only if they have" in "Robots can be truly intelligent only if they have" and use plural in the subsequent statements
*Re-phrase for the sake of readability "As a home service robot, people hope that it can actively and correctly recognize services user required by combining the user behavior with the current scenario";"robot have poor intelligence and universality."; "The robot in a new environment just like a new-born baby, there is no previous experience in its brain, and the process of learning the environment based on reinforcement is also a process of accumulating experience through continuous trial in unfamiliar environment"; "Model the environment with the scenario information that can be recognized by the robot, in which the scenario information conntains the natural environment information, user status information and device switch information"
*delete "flowchat.pdf","table.png" and "result.png"
*unify SARSA and sarsa
*Altorithm
*al -> al.
*badroom
*tempurature
*gotten
*arbitratily
*Altorithm
*Congition
*Processiong
*don’s
*service. the reward
*Performence

Round 2

Reviewer 2 Report

Paper has been improved but a further review round is required.

For example, "The development of IoT technology makes it easier for robots to access environmental information [1], [2], including the on/off and open/closed status of devices and furniture" I don't think that "the on/off and open/closed status of devices and furniture" might be considered and environmental information, but a context information. Similarly, in "Therefore, we focused on applying reinforcement learning to the service cognition of robots in the home environment to provide robots with the ability to learn service cognition by themselves" I don't think that service cognition is something to be learnt, but it is a skill. Avoid editing (1) as formula since it is too much simplycistic. Avoid blank space at the end of page 17. In the conclusion, past tense verbs are necessary as well as a general re-organization since the section has many redundant parts. First and family names are reversed in Reference 10 and then also in the text (at line 56, edit "Claudio et al. [10]" in "Savaglio et al. [10]"). A deep review is necessary to improve the overall style and readability (e.g., " When robots encounter scenarios similar to historical experience, they can make quick decisions based on experience")

Round 3

Reviewer 2 Report

Authors improved the paper but a further review is needed to fix minors ("an context–action", "In an context model") and improve the overall style ("with the current context information obtained by the IoT in a smart home").
Moreover, authors are invited to outline future work and improve Fig.14 quality.
